# Docking for EP4R antagonists active against inflammatory pain

Stefan Gahbauer [1,12], Chelsea DeLeon[2,12], Joao M. Braz [3,12], Veronica Craik [3], Hye Jin Kang [2,10], Xiaobo Wan[1], Xi-Ping Huang[2], Christian B. Billesbølle [1], Yongfeng Liu [2], Tao Che [2,11], Ishan Deshpande[1], Madison Jewell [3], Elissa A. Fink [1], Ivan S. Kondratov[4,5], Yurii S. Moroz [6,7], John J. Irwin [1], Allan I. Basbaum [3] ✉, Bryan L. Roth [2,8,9] ✉ & Brian K. Shoichet [1] ✉

The lipid prostaglandin $E_2$ ($PGE_2$) mediates inflammatory pain by activating G protein-coupled receptors, including the prostaglandin E2 receptor 4 (EP4R). Nonsteroidal anti-inflammatory drugs (NSAIDs) reduce nociception by inhibiting prostaglandin synthesis, however, the disruption of upstream prostanoid biosynthesis can lead to pleiotropic effects including gastrointestinal bleeding and cardiac complications. In contrast, by acting downstream, EP4R antagonists may act specifically as anti-inflammatory agents and, to date, no selective EP4R antagonists have been approved for human use. In this work, seeking to diversify EP4R antagonist scaffolds, we computationally dock over 400 million compounds against an EP4R crystal structure and experimentally validate 71 highly ranked, de novo synthesized molecules. Further, we show how structure-based optimization of initial docking hits identifies a potent and selective antagonist with 16 nanomolar potency. Finally, we demonstrate favorable pharmacokinetics for the discovered compound as well as anti-allodynic and anti-inflammatory activity in several preclinical pain models in mice.

Inflammatory pain signaling is initiated by nociceptors upon tissue damage, by heat, toxins, infections, or mechanical stress. Injured cells release chemical mediators, which collectively generate an "inflammatory soup" including chemokines, cytokines, bradykinin and prostanoids[1]. These mediators activate receptors expressed on the peripheral terminals of nociceptors, lowering their activation threshold for signaling. Consequently, normally innocuous or mildly painful stimuli can provoke or exacerbate pain at the site of inflammation leading to allodynia and hyperalgesia, respectively[2].

Prostanoids are lipid mediators that modulate the activity of peripheral and central nociceptors[3]. Among these, prostaglandin $E_2$ ($PGE_2$) is the major pro-inflammatory mediator inducing pain hypersensitivity by activating prostaglandin E2 receptors (EP1-4), which are members of the family of G protein-coupled receptors (GPCRs)[4]. The current standard of care for inflammatory pain begins with nonsteroidal anti-inflammatory drugs (NSAIDs), such as aspirin or ibuprofen. By inhibiting the COX-1 and COX-2 cyclooxygenases, NSAIDs reduce prostanoid biosynthesis and can counteract tissue

[1]Department of Pharmaceutical Chemistry, University of California San Francisco, San Francisco, CA 94158, USA. [2]Department of Pharmacology, University of North Carolina at Chapel Hill School of Medicine, Chapel Hill, NC 27514, USA. [3]Department of Anatomy, University of California San Francisco, San Francisco, CA 94158, USA. [4]Enamine Ltd., Kyiv, Ukraine. [5]V.P. Kukhar Institute of Bioorganic Chemistry and Petrochemistry, National Academy of Sciences of Ukraine, Kyiv, Ukraine. [6]Chemspace LLC, Kyiv, Ukraine. [7]National Taras Shevchenko University of Kyiv, Kyiv, Ukraine. [8]National Institute of Mental Health Psychoactive Drug Screening Program, University of North Carolina at Chapel Hill School of Medicine, Chapel Hill, NC 27514, USA. [9]Division of Chemical Biology and Medicinal Chemistry, University of North Carolina at Chapel Hill Eshelman School of Pharmacy, Chapel Hill, NC 27514, USA. [10]Present address: Department of Biotechnology, College of Life Science and Biotechnology, Yonsei University, Seoul, South Korea. [11]Present address: Center of Clinical Pharmacology, Department of Anesthesiology, Washington University School of Medicine, St. Louis, MO 63110, USA. [12]These authors contributed equally: Stefan Gahbauer, Chelsea DeLeon, Joao M. Braz. ✉e-mail: allan.basbaum@ucsf.edu; bryan_roth@med.unc.edu; bshoichet@gmail.com

injury-induced inflammatory pain. Common side-effects of NSAIDs, such as gastrointestinal bleeding, are associated with COX-1 inhibition, while newer NSAIDs that selectively inhibit COX-2, such as celecoxib and rofecoxib, can increase the risk for cardiovascular diseases[5,6]. These adverse side effects of NSAIDs arise from their disruption of prostanoid biosynthesis and homeostasis, which ideally balances pro-inflammatory (e.g. $PGE_2$), as well as pro- and anti-thrombotic prostanoids (thromboxane $A_2$ and $PGI_2$, respectively). A more targeted intervention in prostanoid signaling for the treatment of inflammatory pain may therefore lie in the modulation of $PGE_2$ receptor activity[7].

The EP4 receptor (EP4R) mediates $PGE_2$-induced sensitization of peripheral nociceptors and several efforts have sought EP4R antagonists[8–10] (see Supplementary Fig. 1). These produced several clinical candidates[11], e.g. CR-6086 for rheumatoid arthritis[12], LY3127760 for inflammatory pain[13], and BGC-20-1531 for migraine[14,15]. Intriguingly, owing to the contribution of prostaglandins to tumor cell proliferation and survival, EP4R antagonists also entered clinical trials as novel cancer therapeutics, e.g. ONO-4578[16] and E7046[17]. In 2016, the EP4R antagonist Grapiprant (CJ-023423)[18] was accepted by the FDA's Center for Veterinary Medicine for the treatment of osteoarthritis-induced chronic inflammatory pain in dogs, highlighting the potential of EP4R antagonists for chronic pain relief[19]. However, to date, no EP receptor antagonists are approved for pain management in humans.

Empowered by the rapidly increasing availability of experimental GPCR structures, structure-based design of GPCR ligands has successfully discovered new leads to therapeutics[20]. Here, we sought to diversify EP4R antagonist scaffolds using a virtual structure-based approach. Although computational docking against lipid-binding proteins can be challenging[21], recent successes of such methods against several target classes supported the plausibility of the endeavor[22–27]. In this study, we docked a library of roughly 440 million "tangible" make-on-demand molecules against the EP4R orthosteric site. This docking campaign yielded potent antagonists that were effective in nociceptive processing pain in mouse models. Prospects for further development of this strategy and of these compounds will be considered.

## Results

### Computational docking identifies EP4R antagonists

Inspection of the EP4R crystal structure bound to the antagonist ONO-AE3-208[28] demonstrated that the receptor's ligand binding pocket, between the extracellular halves of transmembrane helices TM1 and TM7, is partially exposed to the cell membrane (Fig. 1a). Accordingly, using short coarse-grained and all atom molecular dynamics (MD) simulations (Methods, Supplementary Data 3), we embedded the receptor in a 1-palmitoyl-2-oleoyl-sn-glycero-3-phosphocholine (POPC) lipid bilayer. The coordinates of the hydrophobic carbon tails were subsequently used to assign a low-dielectric layer around the receptor (Supplementary Fig. 2).

The library docking program, DOCK3.7[29], samples ligand orientations around pre-defined hot-spots in the receptor orthosteric site[30], and within each orientation precalculated ligand conformations[31,32]. Each ligand configuration is evaluated for fit using a physics-based scoring function that calculates electrostatic and nonpolar complementarity using a probe-charge model of Poisson-Boltzmann electrostatics and a van der Waals term adapted from the AMBER potential[33], correcting for context-based ligand desolvation using a Generalized-Born formalism[34,35]. The method inevitably under-samples states, especially those owing to protein flexibility and the to the roles of ordered waters, while the scoring function ignores important terms, such as polarizability, underweights others, like hydrophobicity, and struggles to balance polar and non-polar interactions. Accordingly, we have found it prudent to benchmark docking screens by control calculations using property-matched, extrema, and property-unmatched decoy sets[36–38], which challenge our ability to highly rank known

ligands—here EP4R antagonists—against sets of non-ligands that test different parts of the scoring function and docking sampling. Only after we had achieved favorable enrichments and ligand poses against these several decoy sets, did we turn to screening the ZINC15 compound library[39] of neutral and anionic compounds (approximately 400 million compounds), with molecular weight (MWT) between 250 and 500 amu and calculated logP ≥1. After clustering and filtering the top 300,000 scored molecules for diversity and dissimilarity to known EP4R ligands (Methods), we visually inspected the top-ranking 10,000 compound cluster representatives and purchased 40 diverse make-on-demand compounds from Enamine, most or all of which have not previously been synthesized.

This library only contained 28 million anionic compounds (7%), underrepresenting anions versus their prevalence in approved therapeutics[40]. As most reported EP receptor ligands are negatively charged, we sought to expand the number of anions in the library. By searching the Enamine REAL database for substructure patterns of carboxylic acid and several of its bioisosteres (Methods and Supplementary Fig. 3), we found 39 million additional anions (MWT 250–400 amu). After generating 3D conformer libraries (Methods) and docking this additional set of lead-like anionic compounds, we followed a similar hit-picking strategy and selected 37 more compounds. Thirty-one of these 31 (84%) were successfully synthesized by Enamine. We initially screened all 71 docking hits (1–71) against EP4R at 10 μM measuring radioligand competition with $^3H$-$PGE_2$ (Fig. 1b, Supplementary Data 1, and Supplementary Methods). Ten compounds (62–71), that displaced approximately 20% of radio-labeled $PGE_2$ were further tested in functional luminescence-based β-arrestin2 recruitment PRESTO-Tango assays[41]. While the radioligand assay has the advantage of measuring direct binding, the PRESTO-Tango assays are a rapid way to begin to measure functional activity, here receptor antagonism.

Six compounds (62, 65, 66, 68, 70 and 71) showed dose-dependent antagonist activity at EP4R (see Fig. 1a, d) with compound 71 having an $IC_{50}$ of 850 nM. Based on apparent potency and on low similarity to previously described EP4R ligands (ECFP4-based Tanimoto coefficients of 0.27 and 0.23 to EP4R antagonists in the ChEMBL database[42], respectively), both compounds 66 and 71 were prioritized for structure-based optimization.

### Docking hit optimization and structure-activity relationship

As observed in recently solved agonist- and antagonist-bound EP and thromboxane receptor crystal and cryo-EM structures[28,43–48], in the predicted poses, the carboxylates of 66 and 71 hydrogen bond and ion pair to Thr168 (ECL2), Tyr80 (TM2), and Arg316 (TM7), respectively (Fig. 2a, b). Both docking hits have unusual cyclic rings attached to their acidic warhead—for 66 a cycloheptane and for 71, a spiro[3.5] nonane (see Figs. 1d, 2a, b). In contrast, most of the previously reported EP4R antagonists contain benzoic acid or acylsulfonamides (Supplementary Fig. 1)[8,9]. Thr76, a suggested selectivity filter for compounds binding to EP4R over other EP receptors, is modeled to hydrogen bond to the pyrimidine of compound 71 and to the central amide carbonyl of compound 66. Additional hydrogen bonds are predicted between Ser319 (TM7) and the secondary amino group of 71 or the amide of 66. Compared to the co-crystalized antagonist ONO-AE3-208 (Fig. 1a), in their docked poses both antagonists insert deeper into the orthosteric site of the receptor.

We next sought to improve the affinities of the initial docking actives. Here, we docked analog libraries containing 1873 and 13,315 structurally similar compounds to compounds 66 and 71, respectively, from the Enamine REAL database, against EP4R. Virtual analog libraries were generated by separating the initial hit scaffold into its original building block reagents according to Enamine's make-on-demand reaction scheme. For each reagent, decorations and structurally similar building blocks were selected from the Enamine REAL Database

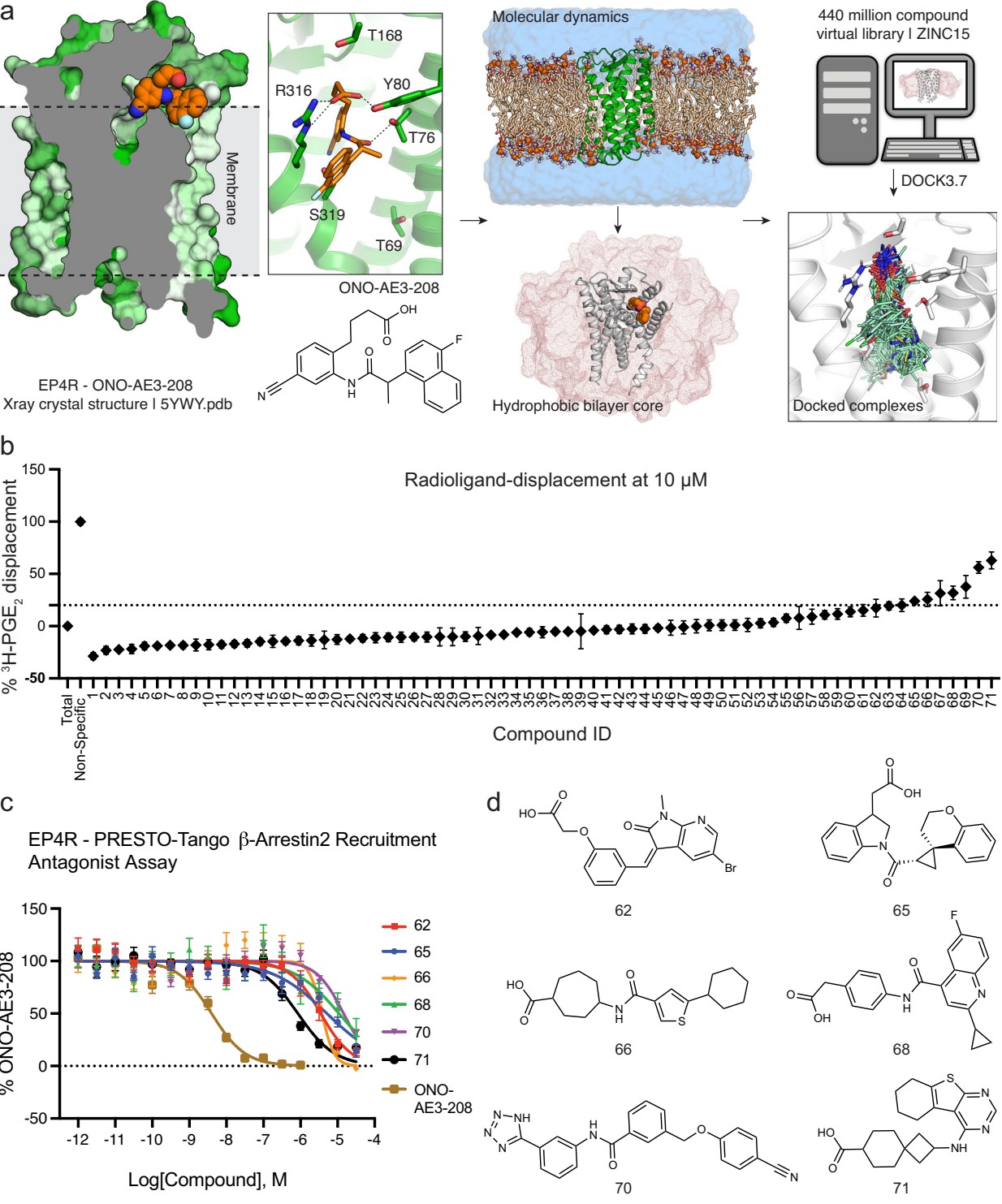

**Fig. 1 | Computational docking and in vitro testing for EP4R antagonists. a** The crystal structure of the EP4R-ONO-AE3-208 complex (PDB 5YWY)[29] was targeted for docking. To reflect the low dielectric of the hydrophobic lipids surrounding the receptor, we equilibrated a POPC lipid bilayer around the protein using molecular dynamics simulations. About 440 million molecules were then docked against the EP4R structure. **b** Displacement of radiolabeled $^3$H-PGE$_2$ by docking hits at 10 μM.

**c** Concentration-response curves of docking hits using a PRESTO-Tango β-arrestin2 recruitment assay. **d** Docking hits demonstrating antagonist activity at EP4R. Data in **b**) represents mean ± SEM of three (for compounds from the ZINC15 library) or four (for compounds from the additional anion library) technical repetitions (see Supplementary Data 1). Data in **c**) represents mean ± SEM of three independent experiments. Source data are provided as a Source Data file.

(RDB) so that the final product analogs could be readily synthesized with the well-characterized RDB reactions. E.g. for 71, decorations to the aminospiro[3.5]nonane-carboxylate building block (EN300-6505077) were combined with decorations of three different halide coupling reagents: 4-chloro-5,6,7,8-tetrahydrobenzo[4,5]thieno[2,3-d]pyrimidine

(EN300-01504), 4-chloro-7-methyl-5,6,7,8-tetrahydrobenzo[4,5]thieno[2,3-d]pyrimidine (EN300-07402) and 4-chloro-6,7-dihydro-5H-cyclopenta[4,5]thieno[2,3-d]pyrimidine (EN300-05102). Based on the docking scores and on visual inspection, 13 analogs of compound 71 and 20 analogs of compound 66 were synthesized (see Supplementary Data 1).

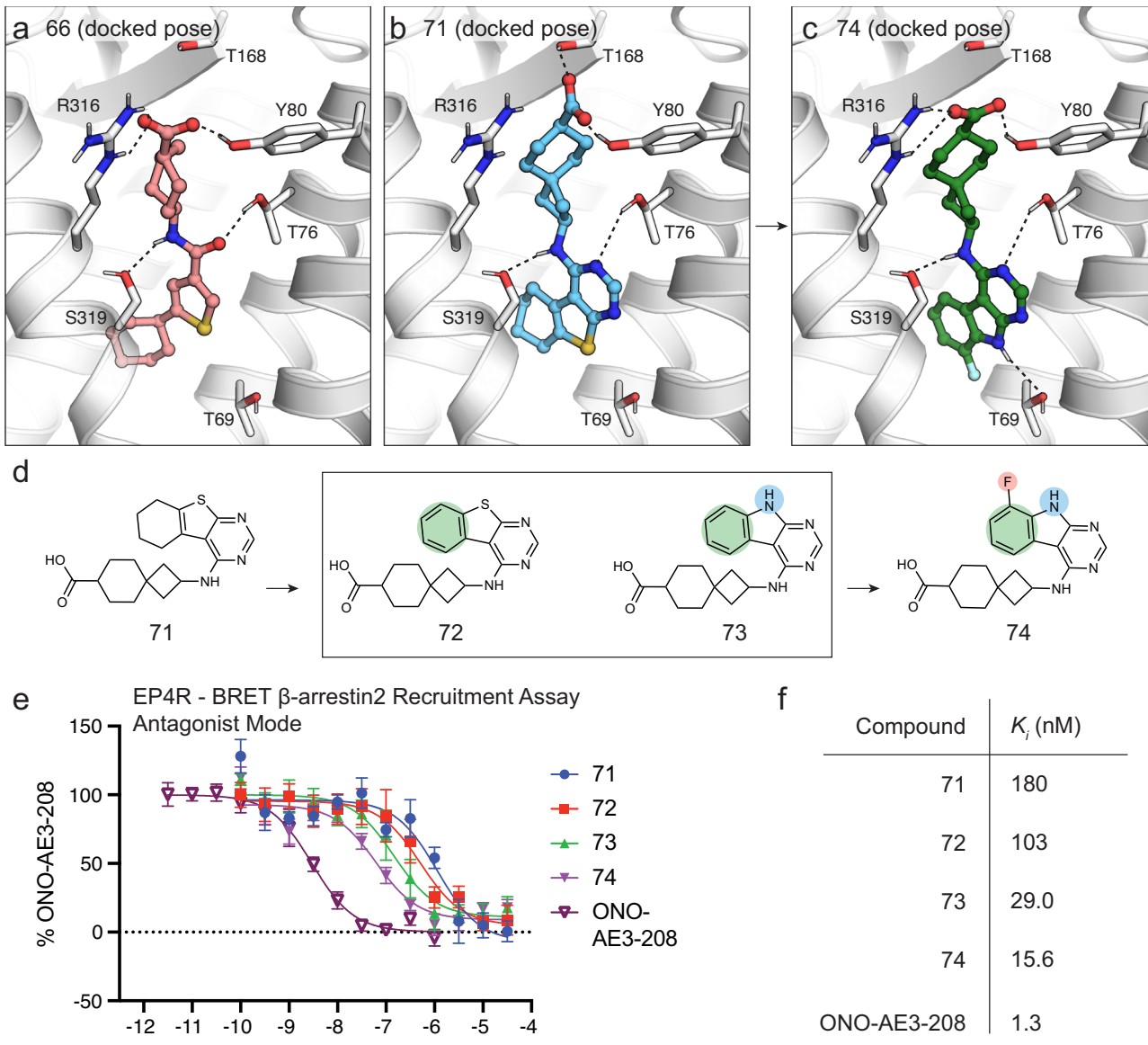

**Fig. 2 | Predicted binding poses of docking hits and their structure-based optimization. a**, **b** Docked poses of 66 and 71, respectively. **c** Docked pose of 74. Dotted lines indicate hydrogen bonds. Note that while the carboxylate of 71 is out of hydrogen-bond range of Arg316, the two groups maintain an ionic interaction, contributing to the overall positive electrostatic potential in this region. **d** Structure-based optimization of 71. **e** Concentration-response curves of compounds shown in **d)** using BRET-based EP4R-arrestin interaction reporter assay. **f** Calculated Ki values (using the Cheng-Prusoff approximation) of compounds shown in **d)**. Data in **e)** represents mean ± SEM of three independent experiments. Source data are provided as a Source Data file.

Although the compound 66 series ultimately did not lead to a highly potent antagonist (at best 32 nM, Supplementary Fig. 4), the compound 71 scaffold was successfully optimized to low nanomolar potency.

Compound analogs were evaluated by employing a bioluminescence resonance energy transfer (BRET) assay using wild-type EP4R attached to Renilla luciferase and a mVenus N-terminally modified human β-arrestin2. Like PRESTO-Tango (above), this assay measures functional antagonism, but since it does so by measuring direct recruitment of effectors to the receptor, it is not subject to the potential downstream amplification issues of PRESTO-Tango. Small changes of the initial 71 scaffold ($K_i = 180$ nM, using the Cheng-Prusoff approximation) showed meaningful improvement in potency (Fig. 2d–f). Exchanging the terminal cyclohexane by benzene (72, $K_i = 103$ nM) moderately improved potency. Furthermore, the replacement of the benzothiophene by an indol group reduced the oxidation liabilities of the series and resulted in the most potent antagonist of the first set of 71 analogs, 73 with a calculated $K_i$ of 29 nM. As the number of non-hydrogen atoms did not differ between 71 and 73, despite potency increasing almost 10-fold, ligand efficiency rose from 0.35 to 0.40 kcal/mol per "heavy" atom. Finally, the addition of a fluorine at the indole C7 position in the second round of scaffold optimization led to compound 74, with a calculated $K_i$ of 15.6 nM (see Fig. 2c, d). The predicted binding pose of 74 revealed formation of an additional hydrogen bond to the deeply buried Thr69 (TM2), an interaction not observed in the crystal structure of EP4R bound to the antagonist ONO-AE3-208 (Figs. 1a, 2c). Analogs 75 and 76 were designed to test the predicted hydrogen bonds between 74 and EP4R by methylating the indole or central amine groups of the lead scaffold and indeed showed up to 10-fold reduced potencies (Supplementary Data 1). Furthermore, the EP4R mutants Thr69Ala and Thr76Ala were generated to corroborate the predicted binding pose of 74 (see

Supplementary Fig. 5). Encouragingly, 74 showed a five-fold reduced potency at the Thr69Ala mutant, supporting the predicted formation of a hydrogen bond of the compound to Thr69. Similarly, the antagonist's potency was reduced 30-fold at the Thr76Ala mutant indicating critical interactions between the compound and the aforementioned selectivity filter for EP4R-specific ligands. The competitive antagonism of 74 was investigated and shown by Schild analysis (Supplementary Fig. 5). In an orthogonal functional cAMP production-based assay, the apparent dissociation constant $K_B$ of 16 nM was calculated for 74 using the Cheng-Prusoff equation (Supplementary Fig. 6). Furthermore, in a radioligand displacement assay using $^3$H-PGE$_2$, 74 had, as expected, a higher $K_i$ value of 366 nM, as pretreatment with the labeled agonist likely induced active-state conformations in the expressed EP4 receptors (Supplementary Fig. 6)[49].

### Pharmacological characterization of compound 74

The most active compound that emerged from structure-based optimization, 74, was subsequently tested against an off-target panel. Using the $^3$H-PGE$_2$ displacement assay, compound 74 showed no binding to EP1, EP2 or EP3 receptors (Supplementary Fig. 6). Furthermore, no off-target activities were observed in functional cAMP production-based assays against the prostaglandin D$_2$ or thromboxane A$_2$ receptors, respectively. Against a PRESTO-Tango-based panel containing 320 GPCRs, we observed no agonist activity against any receptor at 10 μM compound concentration. This panel includes several adverse drug reaction receptors such as the serotonin receptor 5HT2b, M1 and M2 muscarinic receptors and adrenergic receptors. Due to its pyrimido-indole group, compound 74 was also tested against a panel of 97 human kinases, using the KINOMEscan technology (Supplementary Data 2, Supplementary Fig. 7)[50]. At 10 μM, compound 74 showed notable inhibition of five kinases that was confirmed in subsequent dose-response experiments (Supplementary Fig. 7). Most importantly, compound 74 inhibited the bone morphogenic protein receptor type 2 (BMPR2) kinase with a $K_d$ of 30 nM, placing it among the most potent BMPR2 inhibitors reported[51]. Although further study of BMPR2 inhibition by compound 74 is beyond the scope of this work, it may merit further consideration given the potential use of such inhibitors in the treatment of certain cancers, and the potential use of EP4R antagonists to treat the same[52]. Last, we observed no inhibitory activity of 3 μM of compound 74 against either COX-1 or COX-2, nor against the nuclear hormone receptor PPARγ (Supplementary Fig. 7), all of which are relevant to inflammatory signaling pathways. In summary, the EP4R antagonist 74 exhibits a favorable selectivity profile against more than 400 potential off-targets, however, we note that an even more comprehensive safety assessment is merited before considering advancing the compound to clinical candidacy.

### In vivo analysis of EP4R antagonists

To assess the therapeutic potential of compound 74 we first evaluated its solubility in different vehicles (Supplementary Fig. 8) and then investigated its pharmacokinetics after 10 mg/kg intraperitoneal (i.p.) injections in mice (Supplementary Fig. 9). Likely due its carboxylate, compound 74 was peripherally restricted and showed only marginal brain and cerebral spinal fluid exposure; much higher levels were recorded in plasma ($C_{max}$ = 6,670 ng/ml; AUC = 1,450,000 ng*min/ml). When formulated for 10 mg/kg i.p. injections, the sodium salt of 74 (compound 77) had even higher plasma exposure ($C_{max}$ = 9,910 ng/ml; AUC = 2,370,000 ng*min/ml). We also tested two previously reported EP4R antagonists, CJ-42794 and Grapiprant under the same conditions. As for 77, CJ-42794 had high plasma exposure ($C_{max}$ = 20,600 ng/ml, AUC = 2,720,000 ng*min/ml); Grapiprant's exposure was much lower ($C_{max}$ = 834 ng/ml, AUC = 80,600 ng*min/ml). Compared to CJ-42794, compound 77 showed higher concentrations in blood two to four hours after injection. After intravenous (i.v.) injection at 10 mg/kg,

compound 77 reached high blood exposure, with an AUC of 2,770,000 ng*min/ml; oral (p.o.) administration (30 mg/kg) generated an AUC of 6,480,000 ng*min/ml, yielding an oral bioavailability of 78% (Supplementary Fig. 9). Meanwhile, the reported oral bioavailability values of CJ-42794 and Grapiprant are 73% and 5%, respectively[18,53].

Encouraged by its high selectivity and favorable pharmacokinetics, we next tested compound 74 and its sodium salt 77 in both short and longer-term chronic in vivo inflammatory pain models. Carrageenan-induced mechanical allodynia of the hindpaw was significantly reduced by either a 30 mg/kg i.p. or p.o. dose, administered one day after intraplantar injection of 30 μg carrageenan (Fig. 3a). In contrast, in absence of injury, compound 77 had no effect on baseline mechanical thresholds and did not induce motor impairment in the rotarod test (Supplementary Fig. 10). Using the same inflammation model, we observed equal anti-allodynic effects of 10 mg/kg i.p. injected of 77 and CJ-42794 (Fig. 3b), despite the approximately five-fold higher potency of CJ-42794 at EP4R (Supplementary Fig. 6). Higher dosing of CJ-42794 was not possible due to limited solubility in the screened formulations (Supplementary Fig. 8), consistent with its relatively high cLogP (5.5 vs 4.1 for 74). Pretreatment with compound 77 also dose-dependently prevented mechanical hypersensitivity induced by an intraplantar injection of 10 μg of PGE$_2$, 30 min after i.p. administration of the antagonist (Fig. 3c). The reversal of the hypersensitivity at the highest dose suggests that activation of EP4R is a major contributor to the PGE$_2$-induced hypersensitivity. Finally, to determine whether 74 retains its anti-allodynic properties in the setting of a more prolonged inflammation, we turned to the complete Freund's adjuvant (CFA) model. Here, we injected a solution containing 50% CFA into the hindpaw of mice and 3 days later (at peak inflammation), we tested the mechanical thresholds before and 1 h after 74. Figure 3d illustrates that 10 and 30 mg/kg dose of 74 i.p. significantly decreased the CFA-induced mechanical allodynia, compared to vehicle control. However, increasing the dose from 10 to 30 mg/kg did not increase the anti-allodynic effect of 74.

As one of the hallmarks of these inflammatory models is paw edema, using a digital caliper we next measured the thickness of the paw of mice injected with carrageenan or CFA, before and after i.p. injection of 77. One day after intraplantar carrageenan, paw swelling was significantly reduced by 30 mg/kg 77, for at least three hours after treatment (Fig. 3e). Furthermore, daily i.p. dosing of 30 mg/kg 77 on four consecutive days, beginning one day after intraplantar CFA, significantly slowed the development of paw edema, reducing paw thickness by up to 20% versus the control group (Fig. 3f). We conclude that 77 has both anti-inflammatory and anti-allodynic properties.

## Discussion

Given their significant contribution to diverse physiological and pathological processes, EP4R antagonists remain highly pursued for therapeutic development. We sought to explore chemical scaffolds using a structure-based approach to expand upon existing EP4R-targeting chemical matter. Three key points emerge from this study.

First, large library docking and structure-based optimization identified EP4R antagonists with low nanomolar potency at the receptor. It is noteworthy that lipid-binding receptors are considered challenging for structure-based approaches like docking and only few campaigns have succeeded in identifying active chemical matter[21]. The hydrophobic nature of both their binding sites and, correspondingly, the ligands that they recognize can increase false-positive rates as the scoring function gets overwhelmed by unspecific hydrophobic complementarity versus specific polar interactions[38]. Here, care was taken to up-weight polar interactions by modeling the receptor in a lipid bilayer, thereby decreasing the effective dielectric constant in the site and increasing the magnitude of polar interactions, and also by increasing the local dipoles of key resides to improve anion recognition.

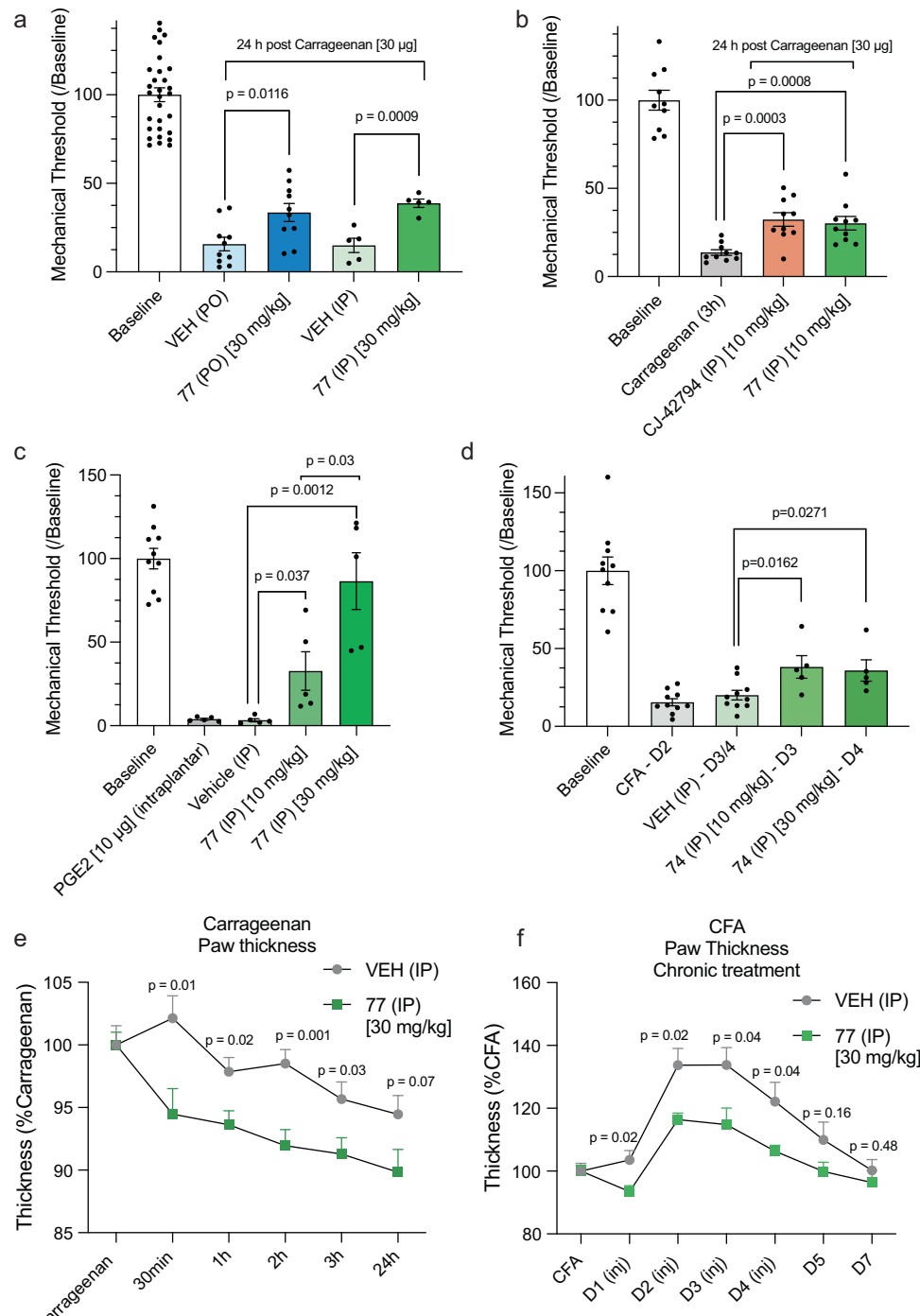

**Fig. 3 | In vivo activity of the discovered EP4R antagonist.** "n" denotes number of independent animals. **a** Carrageenan-mediated mechanical allodynia (von Frey test) is reduced by systemic administration of EP4R antagonists. 77 exhibits significant anti-allodynic properties through i.p. ($n = 5$) or oral (p.o., $n = 10$) administration of 30 mg/kg of the molecule. **b** CJ-42794 and 77 equally reduce carrageenan-mediated mechanical allodynia (von Frey test) upon i.p. injections of 10 mg/kg ($n = 10$ per group). **c** Pre-treatment with 10 or 30 mg/kg 77 by i.p. administration ($n = 5$), 30 minutes prior to intraplantar $PGE_2$ injection, dose-dependently reduces mechanical allodynia (von Frey test). **d** CFA-induced mechanical allodynia (von Frey test) is reduced by i.p. administration of 10 ($n = 5$) and 30 mg/kg ($n = 5$) of 74 after three and four days. **e** Carrageenan-induced paw edema was reduced upon i.p. administration of 30 mg/kg 77 ($n = 10$ per cohort). **f** Injections of 77 on four consecutive days significantly reduced the progression of CFA-induced inflammation ($n = 5$ per cohort). p-values between indicated groups in **a**)-**d**) were calculated with two-tailed unpaired t-tests. p-values in **e**) and **f**) were calculated with multiple unpaired two-tailed t-tests at each timepoint. In **a**)-**f**) data is shown as mean ± SEM. Source data are provided as a Source Data file.

Second, with 26 or 27 non-hydrogen ("heavy") atoms, molecular weight of 350 to 368 amu and clogP 3.9 to 4.1, the discovered antagonists 73 and 74 are small and relatively polar compared to previously known antagonists, such as ONO-AE3-208 (30 heavy atoms, 404 amu, clogP 5.0), BAY-1316957 (33 heavy atoms, 441 amu, clogP 5.8), or Grapiprant (35 heavy atoms, 491 amu, clogP 4.2). These features potentially offer greater opportunity to enhance this scaffold while retaining favorable physicochemical properties. Similarly, although our antagonists are predicted to engage many of the same receptor residues as do previously identified antagonists, they do so

with different groups—the spiro-ring system that supports the warhead carboxylate is without precedent among EP4R antagonists in the public domain, as is the deeply buried hydrogen bond donor within 74's indole group, which appears to interact with Thr69 in the docking calculation (Supplementary Fig. 1). These previously unexplored groups may be useful in the development of future EP4R antagonists and provide structural freedom to operate for this class of molecules.

Third, the most potent compound, 74, exhibited favorable pharmacokinetic properties and bioavailability that facilitated our in vivo analysis of therapeutic potential. We found that systemic delivery of 74 in mice, either by an intraperitoneal or oral route, was anti-allodynic and reduced peripheral inflammatory pain and that both a single injection or chronic administration reduced injury-induced paw swelling.

Certain caveats merit mentioning. Notwithstanding the chemotypes that emerged from the docking, the functionality explored does not confer new EP4R signaling biology, as has occurred in other targets, and our antagonists seem to mostly match the existing clinical candidates in efficacy and PK. Where the discovered molecules do differ from most of the clinical candidates—in their BMPR2 inhibition and their strong peripheral restriction—we are uncertain of the clinical impact. More broadly, how competitive an EP4R antagonist can be versus the well-studied and widely used NSAIDs cannot be known without extensive trials. In principle, attacking downstream signaling, as the EP4R antagonists do, may avoid the pleotropic actions of the COX inhibitors, but potentially at the cost of efficacy. From a technical standpoint, the two most promising docking hits (66 and 71) were both obtained from the relatively small, additionally-generated 39 million anion library, not from the ready-to-dock 400 million ZINC15 set, indicating that library size becomes less relevant if certain chemotypes remain underrepresented (conversely, this illustrates the existence of physical and chemotype holes in even very large libraries, and the importance of filling them[54]). The hit-rate and initial potencies of the EP4R docking hits were low relative to docking campaigns against other GPCRs[54,55], reflecting the recognition and solubility challenges posed by lipid-recognizing receptors and indicating that future improvements of computational tools are required to allow an uniform application of virtual structure-based screening[56]. Finally, although the discovered molecules had convincing anti-allodynia and anti-inflammatory properties in several pain models, administration of relatively high compound doses was required. While this is also true of optimized, investigational drugs that target the EP4 receptor, it does suggest that there is still room for optimization. As recent studies report, co-administration of different NSAIDs via different routes can produce additive and/or synergistic effects[57]. Conceivably, combining EP4R antagonists with therapeutics acting on another target in the inflammatory signaling pathway may offer effective treatment of pain caused by tissue injury-induced inflammation.

These caveats should not obscure the key observations of this study. Docking a large library of tangible molecules identified scaffolds antagonizing the EP4 receptor. Their relatively small size and relatively high polarity enabled optimization to a lead with favorable physical properties, especially for a lipid receptor, and with in vivo activity against inflammatory pain. The strategies for docking against EP4R may be applicable to other lipid-binding receptors and to binding sites exposed to the membrane environment. These molecules and strategies may guide future work to discover EP4 receptor antagonists, which remains an attractive target for inflammation and inflammatory pain, areas for which new modalities are much needed.

## Methods

### Computational modeling and docking

The crystal structure of EP4R in complex with an inhibitory antibody and the antagonist ONO-AE3-208 (PDB 5YWY)[28] was used in the docking calculations. All antibody atoms were removed and both thermostabilizing point mutations, Ala62Leu, Gly106Arg, were converted back to wildtype in UCSF Chimera v.1.14[58]. Input, parameter, and output files from molecular dynamics simulations are provided in Supplementary Data 3.

For embedding the receptor in a POPC lipid bilayer, the unresolved intracellular loop (ICL) 3 was simplified by connecting the open ends on TM5 and TM6 in the experimental structure with an artificial short loop containing Arg215, Gln216 and Ile263 using MODELLER[59]. Next, the apo protein model was converted to the Martini v2.2 coarse-grained model using *martinize.py*[60]. The coarse-grained receptor model was inserted into a POPC bilayer consisting of 295 lipids and 4,726 coarse-grained water beads were added using *insane.py*[61]. The final system contained 8,889 coarse-grained beads in a 10x10x11 nm rectangular simulation box. After 500 steps of steepest-decent minimization, the system was simulated for 50 ns at 310 K with position restraints on all receptor beads using GROMACS 2018[62]. The simulation was performed with a 20 fs timestep, the temperature was kept constant using the v-rescale thermostat with a coupling constant of 1 ps, and the Berendsen barostat was used for constant pressure at 1 bar with a coupling constant of 3 ps in a semi-isotropical manner. Coulomb interactions were treated as reaction field electrostatics with cut-off at 1.1 nm and a relative dielectric constant of 15 was used. The 12-6 Lennard Jones potential was shifted to zero at 1.1 nm. The Martini coarse-grained model was used to efficiently setup and equilibrate protein-lipid systems[63]. Subsequently, the coarse-gained system was converted back to atomistic resolution in the CHARMM36m force field[64] by *backward.py*[65]. The resulting system contained EP4R, 295 lipids and 18,904 water molecules, resulting in a total of 100,742 atoms. The EP4R model was then aligned to the back-mapped receptor structure using PyMOL v.2.1 (www.pymol.org), and the system was minimized for 1,500 steps with a fixed protein structure, followed by 500 steps of minimization of the entire system. The minimized system was further equilibrated in 10 ns atomistic simulation at 310 K applying position restraints on all protein atoms. The simulation was computed with a 2 fs time step using the v-rescale thermostat with a coupling constant of 0.5 ps, and at 1 bar applying the Parrinello-Rahman barostat with a coupling constant of 5 ps in a semi-isotropical manner. The Lennard-Jones potential was switched to zero between 1.0 and 1.2 nm and electrostatic interactions between particles separated by more than 1.2 nm were computed using the particle mesh-Ewald summation. The WT crystal structure in complex with ONO-AE3-208 (excluding the artificial ICL3) was fit to the final snapshot of the atomistic simulation and the coordinates of carbon and hydrogen atoms of the hydrophobic lipid tails of POPC molecules within 17 Å of the receptor (13,066 atoms) were used to assign a low-dielectric (relative dielectric constant $\varepsilon_r = 2$) ring around the protein during docking grid generation (Fig. 1, Supplementary Fig. 2). The unresolved ICL3 in the crystal structure of EP4R should not influence docking scores as it is over 30 Å away from the orthosteric site thus rendering its contribution to the docking score negligible.

For docking calculations with DOCK3.7[29], molecular potential (scoring) grids were generated using the complex containing the receptor and the hydrophobic lipid bilayer core. The receptor structure was protonated using REDUCE[66] and AMBER united-atom charges[67] were assigned (lipid atoms were left uncharged). The partial atomic charges of the sidechain hydroxyl hydrogen atoms of residues Thr76, Tyr80 and Thr168 were increased by 0.4 elementary charge units and the partial charges of corresponding backbone carbonyl oxygen atoms were reduced by the same amount to maintain the net neutral charge of the residues[38]. Electrostatic potentials within the orthosteric pocket were calculated by numerical solution of the Poisson-Boltzmann equation using QNIFFT[68], assigning a relative dielectric constant of 2 to receptor and lipid atoms. In addition, the low-dielectric region of the ligand-binding pocket was extended outwards from the protein surface by 1.9 Å[38]. To compute ligand

desolvation scoring grids with Solvmap[35], the low-dielectric constant was assigned to all receptor and lipid atoms with the volume of the receptor was extended out 0.5 Å from the protein surface[38]. Van der Waals scoring grids were generated with CHEMGRID, excluding lipid atoms. The atomic coordinates of the co-crystalized antagonist ONO-AE3-208 were used to calculate a set of 45 matching spheres representing favorable positions to dock ligand atoms. The docking parameters were optimized by performing control calculations[38] wherein performance was judged by ability to enrich a set of 21 known EP4 receptor antagonists, extracted from the IUPHAR[69] and CHEMBL[42] databases, versus 658 property-matched decoys generated using the DUDE-Z pipeline (Supplementary Fig. 2)[37]. In addition, an "extrema" set of about 315,000 compounds including neutral, negatively and positively charged molecules from the ZINC15 database[39,70] was screened to ensure the model enriches monoanions[37].

For the initial large-scale docking screen, a set of about 400 million compounds, most of which were make-on-demand chemicals from the Enamine REAL database, was downloaded from ZINC15[39]. Molecular weights ranged from 250 to 500 amu, calculated logP values from 1 to 5, and calculated net charges were ≤ −2, −1 and 0. Each library molecule was docked in an average of 11,797 orientations and for each orientation an average of 445 conformations were sampled, amounting to the generation and scoring of more than 746 trillion complexes in 246,535 core hours or 6.8 calendar days on 1500 cores. The 300,000 top-ranking molecules was clustered by ECFP4-based Tanimoto coefficients (Tc) of 0.5 and only the top-scoring member of each cluster was selected to ensure structural diversity among docking hits[22,23]. The resulting 56,002 cluster heads were filtered for novelty by calculating ECFP4-based Tc to 706 EP4R ligands extracted from ZINC15. Molecules with Tc ≥ 0.35 to known EP4R ligands were removed. From the resulting novel 54,734 cluster representatives, the top 10,000 scored molecules were visually inspected, seeking those that not only scored well but did not have liabilities that the docking scoring function can miss, including subtle forms of ligand strain, stranding hydrogen-bond donors in parts of the site without a complementary receptor acceptor[71], and also for their ability to form ionic interactions with Arg316, Tyr80 and Thr168, and to form additional hydrogen bonds to Thr76 and Ser319. Ultimately, 40 compounds were prioritized for synthesis by Enamine, all of which were successfully fulfilled.

A second docking screen of an in-house generated "lead-like"[72] anion library (see below) containing approximately 37 million compounds with molecular weights between 250 and 400 amu from the Enamine REAL database resulted in the generation and scoring of 42 trillion complexes (8857 orientations and 349 conformations, on average, per compound) in 29,157 core hours or 19.4 h on 1500 cores. The top-scored 300,000 compounds were processed as described above, resulting in 26,524 novel cluster heads. The 20,000 top-scored cluster representatives were filtered for intra-molecular strain[32] and the remaining 4175 ligands with torsion energy units (TEU) ≤ 1.5 for the single most strained dihedral angle were visually inspected. Of these, 37 compounds were prioritized for synthesis, of which 32 were successfully synthesized by Enamine (86% fulfillment rate).

### Generation of in-house lead-like anion library
To expand the library of dockable anions, molecules with molecular weight between 250 and 400 amu from the Enamine REAL 2019 release (https://enamine.net/compound-collections/real-compounds)[73] were searched for carboxylates and for 32 carboxylate bioisosteres (Supplementary Fig. 3). By searching through approx. 7.8 billion compounds for SMILES (simplified molecular-input line-entry system) arbitrary target specification (SMART) patterns of the acidic fragments using RDKit (www.rdkit.org), 39.4 million molecules were identified, most of which were carboxylates (61%). The filtered molecules were prepared for docking by generating 3D conformer libraries following the ZINC15/20 building pipeline[22,39,70]. Briefly, protonation states and

tautomers (at neutral pH) were computed with Jchem v.19.15.0 from ChemAxon (www.chemaxon.com), initial 3D models were generated with Corina v.4.2.0026 (www.mn-am.com/products/corina) and conformer libraries were calculated with omega v.2.5.14 by OpenEye (www.eyesopen.com)[74]. Partial atomic charges and desolvation energies were calculated using AMSOL v.7.1 (www.comp.chem.umn.edu/amsol)[35].

### Compound synthesis
All compounds derived from docking and structure-based optimization were synthesized at Enamine. Chemical identities and spectroscopic analysis for active compounds are provided in Supplementary Methods. All initial screening hits and active analogs described in this work were obtained at >95% purity as assessed by LC/MS.

### Cell culture
HTLA cells stably expressing a tTA-dependent luciferase reporter and a β-arrestin2-TEV fusion gene were a gift from the laboratory of R. Axel (Columbia University). HTLA cells were authenticated by morphology, growth characteristics and the successful TANGO assay which demonstrates that both tTA-dependent luciferase reporter and a β-arrestin2-TEV fusion gene are presented in the cells. Cells were maintained at 37 °C and 5% $CO_2$ in DMEM supplemented with 10% FBS, 100 U/mL penicillin, 100 ug/mL streptomycin, 2 μg/mL puromycin, and 100 μg/mL hygromycin B. HEK293T cells were obtained from the American Type Culture Collection (ATCC, CRL-11268), were authenticated by the supplier using morphology and growth characteristics and STR profiling, and tested negative for mycoplasma infection. HEK293T cells were maintained, passaged, and transfected in DMEM medium containing 10% FBS, 100 U ml$^{-1}$ penicillin and 100 μg ml$^{-1}$ streptomycin (Gibco-ThermoFisher) in a humidified atmosphere at 37 °C and 5% $CO_2$. After transfection, cells were plated in DMEM containing 1% dialyzed FBS, 100 U/mL penicillin, and 100 μg/mL streptomycin for BRET assays.

### Radioligand-displacement assay
HEK293T cells stably expressing the wild-type EP4 receptor were used for membrane preparation. Competitive binding assays were performed in 96-well plates in a binding buffer consisting of 25 mM Tris HCl, 10 mM $MgCl_2$, 1 mm EDTA, and 1 mg/mL BSA at pH 7.4. For displacement competition assays, 10 μM of compound was used to compete against a final concentration of 10 nM $^3$H-$PGE_2$ in membrane suspension. For additional competition assays, multiple concentrations of the compound were used to generate a dose-response curves. After the addition of the appropriate materials, plates were incubated at room temperature and in the dark for 90 minutes to allow for equilibration. To terminate the reaction, rapid vacuum filtration was used to transfer the material onto chilled 0.3% PEI filters. The filters were washed with cold 50 mM Tris-HCl, pH 7.4 and read. Results were analyzed using GraphPad Prism 9.0 and the "on-site fit Ki" equation.

### PRESTO-Tango β-arrestin2 recruitment ssay
HTLA cells were plated at $6 × 10^6$ cells per 150 mm cell-culture dish. The following day, cells were transfected with 4 μg of EP4R-tango plasmid using TransIT-2020 (Mirus Bio) according to the manufacturers protocol. On the third day, cells were trypinsized and resuspended in DMEM containing 1% dFBS, 100 U/mL penicillin, and 100 μg/mL streptomycin. The cells were transferred to a poly-L-lysine-coated and rinsed 384-well white, clear-bottomed plate at a cell density of 15,000–20,000 cells per well. At least six hours later, drug solutions were prepared in DMEM containing 1% dFBS, 100 U/mL penicillin, and 100 μg/mL streptomycin. Drug dilutions were added to the appropriate wells for a 3X dilution. For antagonism assays, a 10X solution containing the $EC_{80}$ of $PGE_2$ ($EC_{80}$ = 8 nM) was added fifteen minutes

after the initial compound addition. Cells were incubated with drug overnight at 37 °C and 5% $CO_2$. Following the overnight incubation, the medium and drug solutions were removed and 20 μL of Bright-Glo solution (Promega) was prepared with a 20X dilution in assay buffer (20 mM HEPES and 1× Hank's balanced salt solution, pH 7.4). After 20 minutes, luminescence was counted in a MicroBeta Trilux counter (Perkin Elmer). Relative luminescence units (RLU) were obtained from the experiments and processed in GraphPad Prism. The PRESTO-Tango GPCRome screen was performed according to the protocol previously outlined in Kroeze, et al.[41].

## BRET β-arrestin2 recruitment assay

Renilla luciferase was fused directly to the C-terminus of the EP4 receptor. An mVenus was linked to the N-terminus of human β-arrestin2 by the amino linker SGLRSRRALDS. Together, these components allowed for determination of β-arrestin2 BRET activity at the EP4 receptor. In these assays, HEK cells were plated in a 6-well plate at a density of $0.8$-$1 \times 10^6$ cells per well with DMEM containing 10% FBS, 100 U/mL penicillin, and 100 μg/mL streptomycin. Six hours later cells were transfected with a 1:6 ratio of EP4R-RLuc:mVenus-β-arrestin2 using TransIT-2020 (Mirius Bio) according to the manufacturer's protocol. The following day cells were removed from the 6-well plate with trypsin and transferred to a 96-well white, clear bottomed plate at a cell density of 30,000-35,000 cells per well in DMEM containing 1% dialyzed FBS. Cells were allowed to attach and grow on the plate overnight. For the assay, media was aspirated, and cells were incubated at 37 °C for ten minutes in assay buffer (20 mM HEPES and 1× Hank's balanced salt solution, pH 7.4). Following incubation with assay buffer, 3X of compound was added to each well. For assay determining agonist activity, 5 μM coelentrazine h (Promega) was added to all wells, and plates were read on a PHERAstar FSX Microplate Reader (BMG Labtech) after a 25-minute incubation with compound at room temperature. To assess compounds for antagonism, 4X of the $EC_{80}$ of $PGE_2$ ($EC_{80} = 8$ nM) was added to the wells after the initial 3X addition of compound. Following the addition of the $EC_{80}$ of $PGE_2$, 5 μM coelentrazine h was added to each well. It was found that cells must be incubated at least 25-minutes with the $EC_{80}$ of $PGE_2$ to achieve maximum signal. BRET data was collected from a PHERAstar FSX Microplate Reader using luminescence emission at 485 nm and fluorescent emission at 530 nm. From the experiments BRET, values used for analysis were calculated using a ration of eYFP/RLuc. The net BRET ratio was calculated, normalized to ONO-AE3-203, and fitted using log(inhibitor) versus response in GraphPad Prism 9.0 to display the inhibitory effect of the compound at EP4R. Cheng-Prusoff approximations were calculated using the equation: $K_i = IC_{50}/[(S/K_m)+1]$, where the $IC_{50}$ value was the experimentally determined inhibitory constant of the ligand, S was the $EC_{80}$ value of $PGE_2$, and $K_m$ was the experimentally determined $EC_{50}$ value of $PGE_2$.

## Compound formulations for in vivo studies

Formulation screens for compounds 74, 77, 82, CJ-42794 (MedChem Express, HY-10797) and Grapiprant (MedChem Express, HY-16781) were performed at Enamine biological services Bienta LTD. Compounds were formulated in up to nine different vehicles (Supplementary Fig. 8) at 5 mg/ml and/or lower doses of 2 mg/ml. Solubility was assessed by visual inspection of the formulation using white-light transillumination. Formulations of 74 and 82 in (2-Hydroxypropyl)-β-cyclodextrin (2HPβCD)-saline (20%:80%) required pH manipulation first by adding 2 M NaOH and subsequent neutralization to pH 7 with 1 M HCl. 77 was formulated in 2HPβCD-saline (20%:80%) without adding NaOH but required addition of 1 M HCl to neutralize the formulation. CJ-42794 was formulated in Kolliphor HS15-saline-water (20%:40%:40%) and the solution was neutralized by adding 2 M NaOH. Grapiprant was dissolved in 2HPβCD-saline (20%:80%).

## Pharmacokinetics

Pharmacokinetic studies were performed in accordance with Enamine pharmacokinetic study protocols and Institutional Animal Care and Use Guidelines (protocol number 1-2/2020). CD-1 mice were housed in cages on a standard 12:12 hour light/dark cycle at 22 °C and relative humidity of 40-70%. Each compound was formulated at 2 mg/ml and administered to male 27 CD-1 mice (on average 33 g) via intraperitoneal (i.p) injections reaching a target dose of 10 mg/kg. For the determination of oral bioavailability 77 was dosed 30 mg/kg p.o, (oral) and 10 m/kg i.p. All animals were fasted for hours before dosing. Nine timepoints (5, 15, 30, 60, 120, 240, 360, 480, 1,440 min), each including 3 animals, were chosen for sample extraction. There was one control group animal in each study. Mice were i.p. injected with 150 mg/kg 2,2,2-tribromomethal prior to drawing cerebrospinal fluid (CSF) and blood. CSF was collected using a cisterna magna stereomicroscope and 1-mL syringes. Blood was extracted from the orbital sinus into K3EDTA-containing microtainers. Animals were euthanized by cervical dislocation after collecting blood samples. Blood samples were centrifuged for 10 min at 906 g. Brain samples were extracted from the right lobe and transferred into 1.5 ml tubes after weighing. Immediately after collection, samples were processed, flash-frozen and stored at −70 °C until analysis.

Internal standard (IS) solution (200 μL) was added to plasma samples (40 μL) and after mixing by pipetting and centrifuging (4 min at 5796 g), 0.5 μL of each supernatant was injected into a liquid chromatography-tandem mass spectrometry system (Gradient HPLC system, Shimadzu; MS/MS detector 4000 Q TRAP with TurboIonSpray Electrospray module, MDS Sciex). Internal standard solutions were used to quantify compounds in plasma samples. Brain samples (200 mg ± 1 mg) were homogenized in 800 μL IS solution using zirconium oxide beads (115 mg ± 5 mg) in The Bullet Blender homogenizer for 30 seconds at speed 8 before centrifuging for 4 min at 20,817 g. Supernatant was injected into the LC-MS/MS system. CSF samples (2 μL) were mixed with 40 μL of an internal standard solution by pipetting and centrifuging for 4 min at 5,796 g after which 5 μL of each supernatant was injected into the LC-MS/MS system.

Obtained data of plasma, brain, and CSF samples were conducted at Enamine/Bienta using Analyst 1.5.2 software (AB Sciex). Test compound concentrations below the lower limit of quantification (LLOQ = 10 ng/ml for plasma, 4 ng/g for brain and 1 ng/ml for CSF samples) were designated to zero. Pharmacokinetic data analysis was performed employing noncompartmental, bolus injection or extravascular input analysis models in WinNonlin 5.2 (PharSight). To improve the validity of $T_{1/2}$ calculations, data points below the LLOQ were discarded.

## In vivo behavioral studies

Animal experiments were approved by the UCSF Institutional Animal Care and Use Committee and were conducted in accordance with the NIH Guide for the Care and Use of Laboratory animals (protocol #AN195657). Adult (8-10 weeks old) male C56BL/6 mice (strain # 664) were purchased from the Jackson Laboratory. Mice were housed in cages on a standard 12:12 hour light/dark cycle with food and water ad libitum at 22 °C and relative humidity of 40%.

For all behavioral tests, the experimenter was always blind to treatment. Animals were habituated for 60 min in Plexiglas cylinders before testing. Compound 77 and CJ-42794 were formulated as described above, i.e. in 2HPβCD-saline (20%:80%) or Kolliphor HS15-saline-water (20%:40%:40%), respectively. For 10 mg/kg doses, compounds were prepared in 2.5 mg/ml solutions of which 100 μL were administered to mice (average weight 25 g). For 30 mg/kg doses, 7.5 mg/ml solutions were prepared. Hyperalgesia was induced by a unilateral intraplantar injection of 2% λ-carrageenan (dissolved in saline; Sigma), CFA (50% solution in saline; Sigma) or $PGE_2$ (10 μg/10 μl; Cayman Chemicals). $PGE_2$

was dissolved in ethanol and saline was added to reach the final concentration. The final $PGE_2$ solution contained 1% ethanol. The antihyperalgesic effects of the compounds were tested at 24 h post-carrageenan, 3 (or 4) days post-CFA or 30 min post $PGE_2$. Hindpaw mechanical thresholds were determined with von Frey filaments using the up-down method[75], 60 min after administration of the EP4R antagonists (i.p. or via oral gavage). For the ambulatory (rotarod) test, mice were first trained on an accelerating rotating rod, 3 times for 5 minutes, before testing with any compound. On the test day, latency to fall from the rod was measured 1 h after injection of the compound. The cutoff was 300 s. Statistical tests were performed with GraphPad Prism 9.0 (GraphPad Software Inc., San Diego).

### Reporting summary

Further information on research design is available in the Nature Portfolio Reporting Summary linked to this article.

## Data availability

The ZINC compound library is available to all at https://zinc20.docking.org/ and https://zinc15.docking.org/. The additionally generated anion library is freely available at https://ep4.docking.org/. Most if not all molecules from the anion library are also available in the ZINC-22 compound library (https://cartblanche22.docking.org/). The PDB entry for the EP4R crystal structure used for docking calculations is 5YWY. Figures with associated raw data include Figs. 1, 2 and 3, and Supplementary Figures 4, 5, 6, 7, 9, and 10. All de novo compounds are listed in Supplementary Data 1. Off-target screening results of the lead compound against a panel of 97 human kinases is provided in Supplementary Data 2. Chemical identities, purities (LC/MS), yields and spectroscopic analysis (H-NMR) for active compounds are provided in Supplementary Methods. Input, parameter, and output files of molecular dynamics simulations are provided in Supplementary Data 3. Source data are provided with this paper.

## Code availability

DOCK3.7 is freely available for non-commercial research https://dock.compbio.ucsf.edu/Online_Licensing/dock_license_application.html, and is available online https://blaster.docking.org/.

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

## Acknowledgements

Supported by DARPA HR0011-19-2-0020 (BKS, AIB, JJI & BLR), US NIH R35GM122481 (BKS), US NIH R01GM133836 (JJI), US R35 NS097306 (AIB), Open Philanthropy (AIB), the Facial Pain Research Foundation (AIB).

## Author contributions

S.G., C.D., J.M.B., A.I.B., B.L.R. and B.K.S. conceptualized research. S.G. performed computational modeling and docking screens with input from B.K.S. Virtual anionic compound libraries were generated by S.G. assisted by X.W., Y.S.M. and J.J.I. Ligand optimization was performed by S.G. with input from B.K.S. C.B.B. and I.D. assisted in structural modeling. I.S.K. and Y.S.M. supervised synthesis of compounds at Enamine. X.-P.H., Y.L. and T.C. performed radioligand displacement assays. C.D. performed functional assays assisted by H.J.K. and X.-P.H. X.-P.H. performed off-target screening against the GPCRome panel. S.G. prepared samples for in vivo injections assisted by E.A.F. J.M.B. performed and analyzed the in vivo pharmacology experiments assisted by V.C. and M.J. B.K.S., B.L.R. and A.I.B. supervised the project. S.G. wrote the manuscript with contributions from C.D. and J.M.B., input from all other authors and primary editing by B.K.S., B.L.R. and A.I.B. All authors reviewed the manuscript before submission.

## Competing interests

B.K.S. is founder of Epiodyne, BlueDolphin, and Deep Apple Therapeutics, serves on SABs for Schrodinger LLC, Umbra Therapeutics, Vilya Therapeutics, and consults for Great Point Ventures and for Levator Therapeutics. J.J.I. is founder of BlueDolphin and Deep Apple Therapeutics. Y.S.M. is CEO of Chemspace LLC and scientific advisor at Enamine, Ltd. B.L.R. is founder of Onsero Therapeutics. S.G. and E.A.F. are employed at Deep Apple Therapeutics. The remaining authors declare no competing interest.
