## [Peer Review File · Nature Communications]

REVIEWER COMMENTS

Reviewer #1 (Remarks to the Author):

This paper, entitled "Docking for EP4R Antagonists Active Against Inflammatory Pain," uses a crystal structure model of the prostaglandin EP4 receptor, a lipid receptor in an antagonist-bound state, to screen a vast compound library by in silico The screening was performed. Furthermore, by rationally modifying the derivatives based on the structural information, a novel EP4 antagonist with high affinity and a new backbone was developed. The pharmacological activity of the new compound was confirmed in animal studies and was shown to be equivalent to that of an existing reagent (CJ-42794). To date, no EP receptor antagonists have been approved for human pain management, and the success of this structurally informed drug design should have important implications for drug discovery for GPCRs as well as prostaglandin receptors. It is well summarized and the conclusions are supported by the data.

Lines 148-149: In prostaglandin receptors, it is important that the carboxyl group of the ligand ionically binds to the side chain of an arginine residue that is conserved in the receptor; in EP4, that arginine residue is R316, as shown in the text. In Figures 2a and b, there is no dotted line between the carboxyl groups of 4899 and 4947 and the side chain of R316. Does this mean that they do not interact or that it is not a significant interaction? On the other hand, Fig, 2C shows that 7827_0016 has an ionic bond between the carboxyl group and the side chain of R316. If you believe that R316 is not important for the binding of 4899, 4947 to the receptor, then you should confirm this by calculating the dissociation or inhibition constants from radiolabeled ligand binding experiments or signal activity measurements using mutants such as R316A.

Lines 184-185: The author predicts that 7827_0016 and 4947_7827 can bind Thr69; 7827_0016 and 4947_7827 have improved affinity over 4947_7838, suggesting that this amino acid-ligand interaction is important. Mutants of Thr69 should be produced and confirmed experimentally by radiolabeled ligand binding experiments or by calculation of dissociation and inhibition constants in signal activity measurements.

Reviewer #2 (Remarks to the Author):

The authors have done a nice job docking a large library of tangible molecules to identify novel scaffolds antagonizing EP4R. Lead compounds have decent affinity and anti-inflammatory and analgesic effects are demonstrable in preclinical models. While relatively high doses were required and dose - response

information is limited, the size and polarity of these antagonists compared to existing compounds afford the opportunity for further optimization.

The limitations - at this early stage of development are (i) we do not know if there is a comparative efficacy advantage derived from the structural distinction from published compounds ; (ii) we do not know if other properties - peripheral restriction and BMPR2 inhibition will turn out to be an advantage or otherwise in clinical indications and (iii) we really don't know how sufficiently EP4R antagonism will recapitulate the clinical efficacy of NSAIDs, given the potential redundancy with EP2R or indeed the IPR (was there any competition there?) in some models for example.

Reviewer #3 (Remarks to the Author):

While it might be useful that this work enriched the scaffolds of EP4R antagonists, it is not clear what problems of existing antagonists have and how this enrichment addresses the problems other than providing more candidates for possibility of going into clinic trial. In other words, why are the findings of this work better than others.

Reviewer #4 (Remarks to the Author):

The authors present a computational study focused on the identification of new ligands targeting the EP4 receptor. Through structure-based virtual screening and molecular docking simulations, a set of promising ligands with high affinity for the EP4 receptor were identified. The study highlights their potential as therapeutic agents for various diseases, such as inflammation and cancer. However, further experimental validation and optimization are necessary to advance these ligands towards clinical application.

Authors did a considerable effort, however, I believe revising according to the comments below can improve the manuscript quality.

Comments:

The authors should consider providing a more detailed discussion on the limitations and potential sources of error in their docking methodology. While the optimization and validation steps are commendable, it is important to acknowledge any inherent uncertainties or biases in the approach that may affect the interpretation of the results. Similarly, the experimental validation methods employed in this study are comprehensive; however, the authors should address the potential limitations of each assay technique. It would be valuable to discuss any known drawbacks, potential interferences, or

caveats associated with the radio ligand-displacement assays, PRESTO-Tango β -arrestin2 recruitment assays, and BRET assays to provide a balanced perspective on the experimental results.

The rationale behind the choice of the "lead-like" anion library for the second docking screen could be better justified. It would be beneficial for the authors to elaborate on why this particular library was deemed suitable and how it enhances the chances of identifying novel EP4 receptor ligands compared to other libraries or compound databases.

The manuscript could benefit from a more comprehensive discussion on the potential implications and applications of the identified EP4 receptor ligands. How do these ligands compare to existing therapeutic agents in terms of efficacy, selectivity, and safety?

The manuscript would benefit from improved organization and flow. Some sections could be more clearly connected, and the logical progression of the study could be enhanced. The authors should consider reorganizing the manuscript to provide a smoother and more coherent narrative for the readers.

The authors should address the potential implications of using a simplified intracellular loop (ICL) 3 model and the impact it may have on the docking results and subsequent interpretation.

It is recommended to add additional structural analysis of the identified ligands and their binding modes. Providing insights into key interactions and structural features that contribute to ligand binding and selectivity will further enhance the understanding of the ligand-receptor interactions.

The authors should provide a more thorough discussion on the selectivity of the identified EP4 receptor ligands. While the assays demonstrate their affinity for the EP4 receptor, it is important to address any potential off-target effects or cross-reactivity with other receptors. A comprehensive selectivity profile will strengthen the validity and potential clinical utility of the identified ligands.

The authors should provide a more detailed discussion on the potential toxicity or adverse effects associated with the identified ligands. Addressing the safety profile and potential challenges in minimizing off-target effects or adverse reactions will be crucial for the clinical development of these ligands.

What was the criteria to design the analog library? explain briefly

Captions of each figure are needed to be revised in order to improve the understanding of figures.

Sept 1, 2023

We are grateful to the Reviewers for the time they spent on our manuscript, “*Docking for EP4 Receptor Antagonists Active Against Inflammatory Pain*” by Gahbauer et al. (NCOMMS-23-12775). They were broadly supportive of the work, but did have important critiques. We have been able to address almost all of these, wherever possible taking their suggestions. Doing so has strengthened the manuscript. We take their points in order:

Reviewer 1.

The Reviewer had several kind things to say about the study; we thank them for their generosity and support. Turning to their critiques:

1.1. Lines 148-149: *In prostaglandin receptors, it is important that the carboxyl group of the ligand ionically binds to the side chain of ... R316, as shown in the text. In Figures 2a and b, there is no dotted line between the carboxyl groups of 4899 and 4947 and ... R316. Does this mean that they do not interact or that it is not a significant interaction? On the other hand, Fig. 2C shows that 7827_0016 has an ionic bond between the carboxyl group and ... R316. If you believe that R316 is not important for the binding of 4899, 4947 to the receptor, then you should confirm this by calculating the dissociation or inhibition constants from radiolabeled ligand binding experiments or signal activity measurements using mutants such as R316A.*

We agree that R316 is critical for ligand binding to EP4R, and all top-ranked ligands make ionic interactions with this Arginine. The dotted lines in **Figure 2a-c** indicate hydrogen bonds. In panels **2a** and **2c** hydrogen bonds are shown between the Arg and the carboxylate (in **2a** these were perhaps hard to see, but they were there—we have made them heavier in the revised manuscript).

Still, the Reviewer is correct that such hydrogen bonds are not present in panel **2b**. Instead, in its docked pose the carboxylate of **4947 (71)** has rotated to hydrogen bond with a nearby Thr and a Tyr, the latter of which often participates in concert with the Arg to recognize ligand carboxylates. Notwithstanding this rotation in **2b**, the Arg is still ionically interacting with **4947's (71)** carboxylate, but at 4.5 Å it is out of formal hydrogen-bond range, and so the hydrogen bond is not shown. We now note that the ionic interaction with R316 is maintained among all three compounds in the Figure legend, writing (p.9, lines 172-173):

Note that while the carboxylate of 71 is out of hydrogen-bond range of Arg316, the two groups maintain an ionic interaction, with this residue strongly contributing to the overall positive electrostatic potential in this region of the binding site.

We did also make the R316A mutant receptor; this mutant was no longer activated by PGE2 agonist (Figure S5, and below), preventing determination of an IC₅₀ for the antagonists. This is consistent with the Reviewer's expectation that this is a critical recognition residue. We did also make mutants at two other modeled recognition residues, to which we return below.

	EC50 [nM] (PGE2)	EC80 [nM] (PGE2)
WT	2.2	10
T69A	6.1	30
T76A	210	900
R316A	-	-

1.2. Lines 184-185: The author predicts that 7827_0016 and 4947_7827 can bind Thr69; 7827_0016 and 4947_7827 have improved affinity over 4947_7838, suggesting that this ... interaction is important. Mutants of Thr69 should be produced and confirmed experimentally by radiolabeled ligand binding experiments or by calculation of dissociation and inhibition constants in signal activity measurements.

This point is well-taken. We have now made this mutant, and tested its effect on the IC₅₀ of the lead compound **7827_0016 (74)**. The Thr69→Ala substitution diminishes affinity 5-fold, which is almost exactly the relative difference between '4947_7838 (**72**) (IC₅₀ 103 nM), which does not hbond with this Thr, and '4947_7827 (**73**) (IC₅₀ 29 nM), which does. We do agree that this strengthens the analysis, and thank the Reviewer for the suggestion. We added the dose-response curves of the mutant receptors to Figure S5 and now include a discussion of these mutants in the main text, supporting the docked poses (p. 11 lines 213-218).

Furthermore, the EP4R mutants Thr69Ala and Thr76Ala were generated to corroborate the predicted binding pose of **74** (see **Figure S5**). Encouragingly, **74** showed a five-fold reduced potency at the Thr69Ala mutant, supporting the predicted formation of a hydrogen bond of the compound to Thr69. Similarly, the antagonist's potency was reduced 30-fold at the Thr76Ala mutant indicating critical interactions between the compound the aforementioned selectivity filter for EP4R-specific ligands.

	IC50 [μM] (7827_0016)
WT	0.2
T69A	1.0
T76A	14.7
R316A	-

Reviewer 2.

Reviewer 2 noted that, “the size and polarity of these antagonists compared to existing compounds afford the opportunity for further optimization,” for which we thank them. They also pointed out several of the study’s limitations:

At this early stage of development: (i) we do not know if there is a comparative efficacy advantage derived from the structural distinction from published compounds ; (ii) we do not know if other properties - peripheral restriction and Bmpr2 inhibition will turn out to be an advantage or otherwise in clinical indications and (iii) we really don't know how sufficiently EP4R antagonism will recapitulate the clinical efficacy of NSAIDs, given the potential redundancy with EP2R or indeed the IPR (was there any competition there?) in some models for example.

We agree with these points. What we can say is that by finding genuinely new chemistry and new interactions—in this revision supported by the point-mutants requested by Reviewers 1 and 4—the new antagonists offer new ways forward for EP4R antagonists. More broadly, they illustrate the ability of structure-based large-library docking to reveal these new chemotypes. Nevertheless, the Reviewer’s points are well-taken. We now address them in the manuscript’s penultimate paragraph, writing (p.17, lines 347 - 355):

Certain caveats merit mentioning. Notwithstanding the novel chemotypes that emerged from the docking, the new functionality explored does not confer new EP4R signaling biology, as has occurred in other targets, and the new antagonists seem to mostly match the existing clinical candidates in efficacy and PK. Where the new molecules do differ from most of the clinical candidates—in their Bmpr2 inhibition and their strong peripheral restriction—we are uncertain of the clinical impact. More broadly, how competitive an EP4R antagonist can be versus the well-studied and widely used NSAIDs cannot be known without extensive trials. In principle, attacking downstream signaling, as the EP4R antagonists do, may avoid the pleotropic actions of the COX inhibitors, but potentially at the cost of efficacy.

Reviewer 3

While it might be useful that this work enriched the scaffolds of EP4R antagonists, it is not clear what problems of existing antagonists have and how this enrichment addresses the problems other than

providing more candidates for possibility of going into clinic trial. In other words, why are the findings of this work better than others.

We understand where the Reviewer is coming from. A strength of this work is illustrating how a structure-based approach finds new chemotypes, making new interactions (e.g., the new mutant studies, above and below), for a well-studied target. It also teaches how these might be optimized using make on demand chemistry that is readily available to the community, and how these molecules confer analgesia in vivo. Going from structure -* potent novel chemotypes -* animal activity, competitive with molecules in clinical trials, we think will capture the interest of the readership, as it remains at once a rare result but also one that others can replicate using approaches similar to those used here, and that are openly available to the community.

Still, we agree that a gap is that we haven't found a molecule that beats best-in-class, rather one that meets it. Reviewer 2 raised a similar point and, as mentioned above, we have added caveats to the address that (p.17 lines 347-352):

Notwithstanding the novel chemotypes that emerged from the docking, the new functionality explored does not confer new EP4R signaling biology, as has occurred in other targets, and the new antagonists seem to mostly match the existing clinical candidates in efficacy and PK. Where the new molecules do differ from most of the clinical candidates—in their Bmpr2 inhibition and their strong peripheral restriction—we are uncertain of the clinical impact.

Reviewer 4.

Also found several strengths in the study; we thank them for their support of the work. They did suggest modifications to improve it further:

4.1. The authors should consider providing a more detailed discussion on the limitations and potential sources of error in their docking methodology. While the optimization and validation steps are commendable, it is important to acknowledge any inherent uncertainties or biases in the approach Similarly, ... the authors should address the potential limitations of each assay technique. It would be valuable to discuss any known drawbacks, potential interferences, or caveats associated with the radio ligand-displacement assays, PRESTO-Tango β -arrestin2 recruitment assays, and BRET assays to provide a balanced perspective on the experimental results.

We now discuss the gaps and weaknesses of the methods as the techniques are introduced. For instance, we now write about the docking (p.5, lines 104-116):

The library docking program, DOCK3.7, samples ligand orientations around pre-defined hot-spots in the receptor orthosteric site, and within each orientation pre-calculated ligand conformations. Each ligand configuration is evaluated for fit using a physics-based scoring function that calculates electrostatic and non-polar complementarity using a probe-charge model of Poisson-Boltzmann electrostatics and a van der Waals term adapted from the AMBER potential, correcting for context-based ligand desolvation using a Generalized-Born formalism. The method inevitably under-samples states, especially those owing to protein flexibility and the roles or ordered waters, while the scoring function ignores important terms, such as polarizability, underweights others, like hydrophobicity, and struggles to balance polar and non-polar interactions. Accordingly, we have found it prudent to benchmark docking screens by control calculations using property-matched, extrema, and property-unmatched decoy sets³⁷⁻³⁹, which challenge our ability to highly rank known ligands—here EP4R antagonists—against sets of non-ligands that test different parts of the scoring function and docking sampling. Only after we had achieved favorable enrichments and ligand poses against these several decoy sets, did we turn to screening the ZINC15 compound library⁴⁰...

On p.6, we introduce radioligand competition and Tango, assays, writing (lines 132-138):

We initially screened docking hits against EP4R at 10 μ M measuring radioligand competition with $^3\text{H-PGE}_2$ (**Figure 1B**). Eleven compounds that displaced approximately 20% of radio-labeled PGE_2 were further tested in functional luminescence-based β -arrestin2 recruitment PRESTO-Tango assays.³⁴ While the radioligand assay has the advantage of measuring direct binding, the PRESTO-Tango assays are a rapid way to begin to measure functional activity, here receptor antagonism.

We introduce the use of BRET assays on p.10, writing (lines 195-199):

Compound analogs were evaluated by employing a bioluminescence resonance energy transfer (BRET) assay using wild-type EP4R attached to Renilla luciferase and a mVenus N-terminally modified human β -arrestin2. Like PRESTO-Tango (above), this assay measures functional antagonism, but since it does so by measuring direct recruitment of effectors to the receptor, it is not subject to the potential downstream amplification issues of PRESTO-Tango.

4.2. *The rationale behind the choice of the "lead-like" anion library for the second docking screen could be better justified...*

The simple answer is that we wanted to explore more anions, which we knew EP4 preferred, but which were under-represented in the general ZINC library. In the original ~400 million molecules docked, there were only 28 million anions, 7% of the library, which is much less than their representation among drugs. We thus searched the much larger 2D Enamine library—rather than just the 3D ready-to-dock version of that library—to create new anionic chemotypes to dock. This added another 39 million anions, more than doubling their representation in our dockable library. Ultimately, our two most interesting ligands, **4947 (71)** and **4899 (66)**, emerged from this secondary anion screen. Apart from EP4 and anions, this illustrates the key gaps that still occur in the make-on-demand libraries, notwithstanding their great size, and a way to address them. We address this in the text, writing (p. 5, lines 126-129):

This library only contained 28 million anionic compounds (7%), underrepresenting anions versus their prevalence in approved therapeutics.³³ As most reported EP receptor ligands are negatively charged, we sought to expand the number of anions in the library. By searching the Enamine REAL database for substructure patterns of carboxylic acid and several of its bioisosteres (Methods and **Figure S3**), we found 39 million additional anions (MWT 250-400 amu).

4.3. *The manuscript could benefit from a more comprehensive discussion on the potential implications and applications of the identified EP4 receptor ligands. How do these ligands compare to existing therapeutic agents in terms of efficacy, selectivity, and safety?*

We now write, on p. 17 (lines 352-355):

[...] how competitive an EP4R antagonist can be versus the well-studied and widely used NSAIDs cannot be known without extensive trials. In principle, attacking downstream signaling, as the EP4 antagonists do, may avoid the pleiotropic actions of the COX inhibitors, but potentially at the cost of efficacy.

Apropos of safety, we note that while the compounds are selective on off-target panels that include many adverse drug reaction receptors (e.g., 5HT2b, M1 and M2 muscarinics, adrenergic receptors), the lead has not been through a comprehensive safety assessment, for reasons of scope. We now note this in the manuscript, writing (p. 12, lines 244-247):

In summary, the novel EP4R antagonist **74** exhibits a favorable selectivity profile against more than 400 potential off-targets, however, an even more comprehensive safety assessment is merited before considering advancing the compound to clinical candidacy.

4.4. *The authors should address the potential implications of using a simplified intracellular loop (ICL) 3 model and the impact it may have on the docking results and subsequent interpretation.*

The simplified structure in this area was only used for the membrane-equilibration molecular dynamics simulation wherein the protein structure was positionally restrained to the input crystal structure. For the docking calculations, only the experimentally determined residues were used, i.e. no residues of the unresolved ICL3 region. The lack of the ILC3 region should have negligible docking effects, as it is >30 Å from the orthosteric site, and energy terms fall off steeply with distance. We now write (p. 19, lines 401-408):

The WT crystal structure in complex with ONO-AE3-208 (excluding the artificial ICL3) was fit to the final snapshot of the atomistic simulation [...]. The unresolved ICL3 in the crystal structure of EP4R should not influence docking scores as it is over 30 Å away from the orthosteric site thus rendering its contribution to the docking score negligible.

4.5 ... *add additional structural analysis of the identified ligands and their binding modes. Providing insights into key interactions and structural features that contribute to ligand binding and selectivity will further enhance the understanding of the ligand-receptor interactions.*

This is well-taken, and we have now tested the lead compound **7827_0016 (74)** against mutant receptors that perturb key interactions found in the docked structures, including Arg316-*Ala (perturbing a key modeled ion-pair), Thr69-*Ala (perturbing a putative hydrogen bond with the pyrrole-like nitrogen of the antagonist) and Thr76-*Ala (perturbing a putative hydrogen bond with pyrimidine portion of the antagonist). While the Arg316-*Ala eliminated measurable agonism, and so prevented us from measuring effects on antagonism, Thr69-*Ala and Thr76-*Ala reduced the IC₅₀ of the lead by 5-fold and 30-fold, consistent with—though obviously not proving—the modeled interactions (graph below). We added the following discussion about this new data to the manuscript (p. 11, lines 213-218):

Furthermore, the EP4R mutants Thr69Ala and Thr76Ala were generated to corroborate the predicted binding pose of **74** (see **Figure S5**). Encouragingly, **74** showed a five-fold reduced potency at the Thr69Ala mutant, supporting the predicted formation of a hydrogen bond of the compound to Thr69. Similarly, the antagonist's potency was reduced 30-fold at the Thr76Ala mutant indicating critical interactions between the compound the aforementioned selectivity filter for EP4R-specific ligands.

4.6. *The authors should provide a more thorough discussion on the selectivity of the identified EP4 receptor ligands. While the assays demonstrate their affinity for the EP4 receptor, it is important to address any potential off-target effects or cross-reactivity with other receptors. A comprehensive selectivity profile will strengthen the validity and potential clinical utility of the identified ligands.*

We agree. The lead has been tested against 320 GPCRs and a panel of 97 kinases. The only substantial off-target effect was against BMPR2, which is not a classic toxicity target. We now strengthen this point, writing (p. 12, lines 244-247):

In summary, the novel EP4R antagonist **74** exhibits a favorable selectivity profile against more than 400 potential off-targets, however, an even more comprehensive safety assessment is merited before considering advancing the compound to clinical candidacy.

4.7. *The authors should provide a more detailed discussion on the potential toxicity or adverse effects associated with the identified ligands. Addressing the safety profile and potential challenges in minimizing off-target effects or adverse reactions will be crucial for the clinical development of these ligands.*

While we agree that checking for toxicity is crucial, we believe that going beyond the extensive off-target screens already conducted (**4.6**) would be beyond scope. In the revised manuscript, we do now note the importance of deeper safety pharmacology (**4.3**, above).

4.8. *What was the criteria to design the analog library? explain briefly*

This point is well-taken. We included a more detailed description of analog library design (p. 9, lines 181-190):

Virtual analog libraries were generated by separating the initial hit scaffold into its original building block reagents according to Enamine's make-on-demand reaction scheme. For each reagent, decorations and structurally similar building blocks were selected from the Enamine REAL Database (RDB) so that the final product analogs could be readily synthesized with the well-characterized RDB reactions. E.g. for **71**, decorations to the aminospiro[3.5]nonane-carboxylate building block (EN300-6505077) were combined with decorations of three different halide coupling reagents: 4-chloro-5,6,7,8-tetrahydrobenzo[4,5]thieno[2,3-d]pyrimidine (EN300-01504), 4-chloro-7-methyl-5,6,7,8-tetrahydrobenzo[4,5]thieno[2,3-d]pyrimidine

(EN300-07402) and 4-chloro-6,7-dihydro-5H-cyclopenta[4,5]thieno[2,3-d]pyrimidine (EN300-05102).

4.9. *Captions of each figure are needed to be revised ... to improve the ... figures.*

We have gone through the figure legends, seeking to improve them for clarity.

In summary, we have addressed each of the critiques of the Reviewers, in almost all cases adopting their suggestions for modifications, including making several mutant proteins to interrogate docking-suggested interactions. Together, these have strengthened the manuscript; we thank the Reviewers for their time.

REVIEWERS' COMMENTS

Reviewer #1 (Remarks to the Author):

The authors have satisfactorily answered my questions and appropriately addressed my concerns with additional data.

Reviewer #2 (Remarks to the Author):

My concerns have been addressed by the tempered discussion.

Reviewer #4 (Remarks to the Author):

In the revised manuscript, authors have successfully answered the comments.